

**Future shift of the relative roles of precipitation and temperature in**
**controlling annual runoff in the conterminous United States**
Kai Duan[1], Ge Sun[2], Steven G. McNulty[2], Peter V. Caldwell[3], Erika C. Cohen[2], Shanlei Sun[4],
Heather D. Aldridge[5], Decheng Zhou[4], Liangxia Zhang[4], and Yang Zhang[1]
[1] Department of Marine, Earth, and Atmospheric Sciences, North Carolina State University,
Raleigh, NC, USA
[2] Eastern Forest Environmental Threat Assessment Center, USDA Forest Service, Raleigh, NC,
USA
[3] Coweeta Hydrologic Laboratory, USDA Forest Service, Otto, NC, USA
[4] Key Laboratory of Meteorological Disaster of Ministry of Education, Nanjing University of
Information Science & Technology, Nanjing, Jiangsu, China
[5] State Climate Office of North Carolina, North Carolina State University, Raleigh, NC, USA
Correspondence to:
Kai Duan (kduan@ncsu.edu); Ge Sun (gesun@fs.fed.us)



**Abstract** This study examines the relative roles of climatic variables in altering annual
runoff in the conterminous United States (CONUS) in the 21$^{st}$ century, using an
ecohydrological model driven with historical records and future scenarios constructed
from 20 Coupled Model Intercomparison Project Phase 5 (CMIP5) climate models. The
results suggest that precipitation has been the primary control of runoff variation during
the latest decades, but the role of temperature will outweigh that of precipitation in most
regions if future climate change follows the projections of climate models instead of
the historical tendencies. Besides these two key factors, increasing humidity is
projected to partially offset the additional evaporative demand caused by warming and
consequently enhance runoff. Overall, the projections from 20 climate models suggest
a high degree of consistency on the increasing trends in temperature, precipitation, and
humidity, which will be the major climatic driving factors accounting for 43%~50%,
20%~24%, and 16%~23% of runoff change, respectively. Spatially, while temperature
rise is recognized as the largest contributor in most of the CONUS, precipitation is
expected to be the dominant factor driving runoff to increase across the Pacific Coast
and the Southwest. The combined effects of increasing humidity and precipitation may
also surpass the detrimental effects of warming and result in a hydrologically wetter
future in the East. However, severe runoff depletion is more likely to occur in the
Midwest and South-Central.



# 1 Introduction

Precipitation and temperature are the two key climatic variables that control land water balances and thus control water availability for both ecosystem and humans (Lutz et al., 2014;Milly et al., 2005;Seager et al., 2013;Piao et al., 2010). Changes in temperature interact with changes in precipitation and cause profound shifts in hydrologic paradigms, such as snowpack melting and accumulation (Barnett et al., 2005;Zhang et al., 2015), intensification of hydrologic cycle (Creed et al., 2015;Davis et al., 2015), precipitation partitioning (Duan et al., 2016b;Zhou et al., 2015), extreme floods and droughts (Duan et al., 2016a;Trenberth et al., 2014;Duan and Mei, 2014b), and can lead to hydrological 'nonstationarity' (Milly et al., 2008).

Surface and subsurface (shallow aquifers) runoff is a critical source of fresh water that human populations sustainably have access to (Vörösmarty et al., 2000). The impacts of temperature and precipitation changes on the magnitude and variability of runoff (Ficklin et al., 2009;Arnell and Gosling, 2013;Nash and Gleick, 1991;Vano et al., 2012) have drawn particular attention due to its importance for water supplies. Future changes in precipitation, evaporation, and plant water use are direct driving forces of runoff generation. Climate change alters both precipitation and the partitioning of precipitation into evapotranspiration ($E_T$) and runoff since a warmer climate generally provides more energy for water fluxes between the land and the atmosphere. Although an increase in precipitation may cause increase in both $E_T$ and runoff, the enhanced evaporative demand can result in decreases in runoff efficiency (ratio of runoff to precipitation) (McCabe and Wolock, 2016). Both observation and simulation studies in





the U.S. suggest that higher $E_T$ induced by rising temperature is unlikely to be
counterbalanced by the increase in precipitation and lead to less runoff at large scales
(Duan et al., 2016b;Jackson et al., 2005;Duan et al., 2017). Conversely, warming may
also cause precipitation decrease in some regions and exacerbate the effects of
temperature on runoff change.

6        Several studies have examined the relative contributions of historical changes in

precipitation and temperature to runoff variation at watershed (Karl and Riebsame,
1989), regional (Ryberg et al., 2014;Gupta et al., 2015), and continental (McCabe and
Wolock, 2011) levels across the conterminous U.S. (CONUS). These studies all agree
that precipitation, instead of temperature, explains most of the long-term change and
variability in runoff during the past century. McCabe and Wolock (2011) suggested that
the effects of temperature on runoff may become more substantial under a warming
climate. However, no study in the literature has rigorously investigated the potential
changes in the roles of precipitation and temperature under future climate scenarios.
According to the Parameter-elevation Relationships on Independent Slopes Model
(PRISM) dataset (http://prism.oregonstate.edu/) (Daly et al., 2008), the rate of decadal
change of temperature over the CONUS has reached -0.03~+0.28 °C since 1960s. The
rate of warming is likely to accelerate under intermediate or high emission scenarios
and increase the pressure of water scarcity in many regions in this century (IPCC,
2014;Schewe et al., 2014). In addition, future change in climate is projected to vary
spatiotemporally in both direction and magnitude in the CONUS (Mearns et al., 2012),
thus sensitivity of water budget to climate change may be discrepant across time and



space. Although the possible underestimation of the influence of temperature in altering

regional water resources has been discussed recently (Sospedra‑Alfonso et al.,

2015;Woodhouse et al., 2016), a comprehensive evaluation under different climate

backgrounds and land-cover compositions is still lacking.

We aim to address two questions: (1) to what extent, if any, will the relative roles of

precipitation and temperature in controlling runoff shift if future climate changes follow

the projections of climate models, instead of the tendencies documented in the recent

decades, and (2) how will runoff change in the future and what are the potential roles

of other climatic driving forces besides precipitation and temperature? In the remainder

of the paper, we first describe the methodology of runoff simulation and sensitivity

assessment, and the hydro-climatic datasets used, followed by the results. Then, the

advantages, limitations, and implications of this study are discussed and the conclusions

are drawn.

## 2 Methods

### 2.1 Runoff modeling

The runoff responses to climate change and variability are modeled with the Water

Supply Stress Index model (WaSSI) for 2,099 8-digit Hydrologic Unit Code (HUC-8)

watersheds (http://water.usgs.gov/GIS/huc.html) across the CONUS. WaSSI is a water-

centric ecohydrological model that simulates the land-cover specific water and carbon

cycles on a monthly basis (Caldwell et al., 2012;Sun et al., 2011b). The model

incorporates several mathematical sub-models to describe monthly hydrologic

processes from precipitation input to streamflow routing. A conceptual snow sub-model





(McCabe and Markstrom, 2007) is used to partition the total precipitation into rainfall
and snowfall, and to estimate snowpack melt/accumulation and snow water equivalent
with concern of the mean elevation, latitude, and air temperature in the watershed. $E_T$
is calculated with an ecosystem $E_T$ model developed from the empirical relationships
between $E_T$ and precipitation, potential evapotranspiration (PET), and leaf area index
(LAI) (Sun et al., 2011a;Sun et al., 2011b). These $E_T$ functions were established for 10
different land-cover classes independently to account for the different water demand
within different vegetation, ranging from cropland, deciduous forest, evergreen forest,
mixed forest, grassland, shrubland, wetland, open water, urban area, to barren land.
Then, this $E_T$ estimation is further constrained by soil water availability, which is
simulated using the algorithms of Sacramento Soil Moisture Accounting model (SAC-
SMA) (Burnash, 1995), as well as the processes of infiltration and runoff generation at
monthly basis.
Necessary inputs for WaSSI include monthly precipitation, air temperature, PET, LAI,
and land-cover composition. In this study, the spatial distribution of LAI and the 10
land-cover classes (Fig. 1a) were assumed to be static over time. Monthly climate data
were first scaled to watersheds by the area-weighted averages. All the water balance
components were calculated independently for each land cover class within each
watershed, and then were aggregated monthly means. The model parameters were
acquired from several previous studies, including: (1) The parameters of snow sub-
model were estimated for each Water Resource Region (WRR, i.e., 2-digit HUC
watershed) (Fig. 1b) by comparing regional monthly mean snow water equivalent to





remotely sensed values from the Snow Data Assimilation System (McCabe and
Markstrom, 2007;Caldwell et al., 2012). (2) The parameters of $E_T$ sub-model were
estimated by empirical relationships derived from eddy covariance or sapflow
measurements at multiple sites (Sun et al., 2011a;Sun et al., 2011b). (3) SAC-SMA
parameters used to drive the soil water balance sub-model were developed from soil
physical characteristics documented by the State Soil Geographic Database
(http://soildatamart.nrcs.usda.gov) (Anderson et al., 2006;Koren et al., 2003).

8       The WaSSI model has been validated against observations at U.S. Geological Survey

(USGS) gauged sites at the levels of both 8-digit (Caldwell et al., 2012) and 12-digit
HUC watersheds (Sun et al., 2015b). We here verify the model performance at WRR
and continental scales to complement to previous validations. The simulated annual
runoff, driven by monthly precipitation and temperature from the PRISM dataset, was
compared against the USGS measurements over the entire CONUS (Fig. 2a&2c) and
in the 18 WRRs (Fig. 2b&2d) for the time period of 1961-2010. Despite a slight
overestimation of the minimums, WaSSI shows reliable accuracy in capturing annual
runoff at both CONUS and WRR scales, with R-square statistic reaching 0.91 and 0.95,
and Root Mean Squared Error (RMSE) limited to 29 and 55 mm yr$^{-1}$, respectively.
**2.2 Quantifying the independent effects of climatic variables**
Large-scale water balance can be described as runoff ($R$) equals precipitation ($P$) minus
$E_T$ and changes in soil moisture ($S_M$) and the hydrologically connected snowpack ($S_P$):
$$R = P - E_T + \mathrm{d}S_M/\mathrm{d}t + \mathrm{d}S_P/\mathrm{d}t \qquad (1)$$
While $P$ is the primary water input, changing temperature ($T$) and other climatic factors



interact with each other and affects $R$ by altering the melt/accumulation of snowpack
and controlling $E_T$ with the constraints of vegetation and soil moisture.

3       Here we developed a simple approach of sensitivity test to examine the relative roles

of climatic variables in $R$ variation, as:
$$\Delta R = \sum_{i=1}^{N} E_{Ci} + E_{Int} \quad\quad\quad (2)$$
where $\Delta R$ denotes the change in $R$, which equals the combined effects of variations in
all the climatic variables. $\Delta R$ can be decomposed into the independent effects of each
driving factor ($E_{Ci}$) and the effect of interactions among these variables ($E_{Int}$). $\Delta R$ is
quantified by $R$ change (%) from pre-change period ($t_1$) to post-change period ($t_2$)
driven by changes in all the factors, as $R(C1_{t2}, C2_{t2}, \ldots, CN_{t2})$ −
$R(C1_{t1}, C2_{t1}, \ldots, CN_{t1})$; while $E_{Ci}$ is estimated by $R$ change driven by changes in $Ci$
only, as $R(C1_{t1}, \ldots, Ci_{t2}, \ldots, CN_{t1}) - R(C1_{t1}, \ldots, Ci_{t1}, \ldots, CN_{t1})$. $E_{Int}$ is calculated as
the difference between $\Delta R$ and $\sum_{i=1}^{N} E_{Ci}$, representing the changes in $R$ that cannot be
accounted for by the independents effects. Given that the driving factors may cause
either positive or negative effects on $R$, their contributions are quantified by the relative
weights, as
$$C(Ci) = 100 \times |E_{Ci}| / (\sum_{i=1}^{N} |E_{Ci}| + |E_{Int}|) \quad\quad\quad (3)$$
**2.3 Modeling experiments**
**2.3.1 Climate projection**
Climate projections statistically downscaled from 20 Global Climate Models (GCMs)
(Table 1) of the fifth phase of the Coupled Model Inter-comparison Project (CMIP5)
for both historical forcings and future Representative Concentration Pathways (RCPs)
(the MACAv2-LIVNEH dataset, available at http://maca.northwestknowledge.net/)



were used to test the potential future changes in $R$. RCP4.5 and RCP8.5 were adopted
as representatives of the intermediate and high emission scenarios respectively, which
correspond to radiative forcing of approximately 4.5 W m$^{-2}$ and 8.5 W m$^{-2}$ in 2100
(equivalent to 650 ppm and 1370 ppm $CO_2$) (Moss et al., 2010;IPCC, 2014). The used
climatic variables include monthly $P$, maximum and minimum $T$, solar radiation ($Rs$),
wind speed ($Ws$), and specific humidity ($Sh$) spanning from 1950 to 2099 (Fig. 3).

7       To evaluate the $R$ responses to various changes in future climates, we conducted four

30-year simulation experiments: (i) RCP4.5/2030s (S1 scenario) — near future 2020-
2049 under RCP4.5; (ii) RCP4.5/2080s (S2) — far future 2070-2099 under RCP4.5;
(iii) RCP8.5/2030s (S3) — near future 2020-2049 under RCP8.5; (iv) RCP8.5/2080s
(S4) — far future 2070-2099 under RCP8.5. These four future scenarios cover two post-
change time periods (2030s and 2080s) and are compared to a pre-change period of
1970-1999 (1980s) that represents the baseline level. Traditional sensitivity test
methods usually assume a fixed amount of change (Karl and Riebsame, 1989) or allow
one (or more) of the variables to remain constant over time (McCabe and Wolock, 2011).
In this study, the 30-year-long continuous climate series were used to examine the long-
term patterns while implicitly incorporating the inter- and intra-annual variations. This
large set of climate projections was pooled to enable a robust quantification of the major
uncertainties from GCM structure and emission scenario.
**2.3.2 PET estimation**
Hamon's PET equation has been used for PET estimation in previous WaSSI
simulations because it only requires mean temperature as input and has shown reliable



correlation with actual $E_T$ in historical periods (Lu et al., 2005;Vörösmarty et al., 1998).
Essentially, temperature-based methods perform well because $T$ is correlated with
radiation and humidity at monthly timescale (Sheffield et al., 2012). Such correlations
are the physical bases of the empirical $E_T$ functions, through which variability in $P$, $T$,
and LAI was able to explain the main controls of evaporation and transpiration fluxes
without including the radiative and aerodynamic variables. However, recent studies
revealed that the bias in temperature-based methods could be amplified in future
scenarios of global warming, and led to overestimation of PET, and ultimately $E_T$ and
the severity of surface drying (Milly and Dunne, 2011;Sheffield et al., 2012). Penman-
Monteith (PM) reference $E_T$ (Allen et al., 1998), as a commonly used alternative PET
model, incorporates the effects of surface temperature, humidity, wind, and radiation,
and is considered the most reliable PET approach where sufficient meteorological data
exist (Kingston et al., 2009;Feng and Fu, 2013).

14       In this case, using Hamon equation would lead to 130 mm yr$^{-1}$ larger PET increase

from the baseline to RCP8.5/2080s than that using PM equation (Fig. 4). We assume
that the PM PET projections are more reasonable because the effects of future changes
in $Rs$, $Ws$, and $Sh$ are included as well as $T$. We will focus on analyzing the $R$ changes
and the independent effects of five climatic variables (i.e., $P$, $T$, $Rs$, $Ws$, and $Sh$) based
on PM PET in the remaining of this paper. Effects of $P$ and $T$ evaluated from simulations
of Hamon PET will also be investigated to address the consistency and discrepancy
caused by using different PET methods.





## 3. Results

### 3.1 Projected changes in *R*

Changes in mean annual *R* under future climate change scenarios vary among HUC-8 watersheds (Fig. 5) and WRRs (Fig. 6) across the CONUS. Runoff depletion is projected to cover most part of the Midwest and South-Central U.S. across WRR7~WRR12, with largest decreases over 50% found in WRR10 (Missouri) under RCP8.5. Increases are mainly projected in the Southwest, the north of WRR10, and regions along the Atlantic Coast and Pacific Coast. Extreme increases over 100% are projected in several arid watersheds in WRR15 (Lower Colorado) and WRR16 (Great Basin). However, this may be caused by the inability of GCMs in reproducing the low *P* values in these extremely dry areas. Although the general spatial patterns appear to be similar in the four scenarios, there is an evident expansion of the areas showing either extreme increasing or decreasing trend from 2030s to 2080s under both RCP4.5 (Fig. 5a-5b) and RCP8.5 (Fig. 5c-5d) scenarios.

The large variability of regional changes in *R* (Fig. 6) indicates considerable uncertainties from GCM structure. In most cases, the uncertainty range is limited to -30% ~ +30%, showing both positive and negative changing signals. The distributions of the median lines and Inter-Quartile Ranges (IQRs) suggest a hydrologically drier future in WRR7~12 and WRR14 (Upper Colorado), where consistent decreasing signal is found in all the scenarios. Stronger increasing trend can be found in WRR1 (New England), WRR2 (Mid-Atlantic), WRR17 (Pacific Northwest), and WRR18 (California). Generally, the uncertainty ranges tend to increase from 2030s to 2080s





under both RCPs, and reach a particularly high level under RCP8.5/2080s. There is a
noticeable consistency in this pattern that the GCMs agree more on the simulations in
2030s while the uncertainty aggregates over time toward 2080s, which implies the
limitation of the state-of-the-art GCMs in predicting farther future.
**3.2 Independent effects of climate variables**
The changes in $R$ discussed above are under the combined impact of changing $P$, $T$, $Rs$,
$Ws$, and $Sh$. The independent effects of these factors over the entire CONUS are
illustrated in Fig.7a-7b. $P$ and $T$ are clearly the two most influential factors, which are
projected to cause divergent changes in $R$ due to the increase in $P$ (+15 ~ +31 mm yr$^{-1}$)
and $T$ (+1.8 ~ +5.3 ℃). The median values show that annual $R$ under the independent
$P$ effect is expected to increase by 13 mm yr$^{-1}$ (4%) in 2030s and 24 mm yr$^{-1}$ (8%) in
2080s under RCP4.5, and by 21 (7%) and 30 (10%) mm yr$^{-1}$ at the same time under
RCP8.5. In contrast, the independent effects of $T$ reach -32 (-11%), -50 (-17%), -34 (-
12%), and -80 (-28%) mm yr$^{-1}$ in the scenarios S1~S4. The negative effect of rising $T$
is expected to exceed the positive effect of increasing $P$ and lead to overall decrease in
$R$. However, $Sh$, the third largest contributor, will enhance $R$ by 3%~12% and largely
offset the $T$ effects. Significant increasing trend in $Sh$ is projected under both RCP4.5
and RCP8.5 (Fig. 3e), which will suppress vapor pressure deficit and thus partially
counterbalance the increasing evaporative demand caused by warming. Meanwhile, the
effects of $Rs$ (slightly negative), $Ws$ (slightly positive), and interactions among the
factors ($Int$) are relatively minimal (<3%), suggesting that the variations in $T$, $P$, and $Sh$
can explain the major changes in $R$.



It is worth noticing that much larger uncertainty ranges can be found in the *P* effects.
Compared to the highly consistent increases in *T* and *Sh*, the 20 GCMs constantly
disagree on the changing direction of *P*. Under RCP8.5/2080s, the multi-model result
of *P* effect ranges from -11% to 24%, and the IQR also reaches the highest level (13%).
It indicates that uncertainty in *P* projection is still the largest contributor to the
uncertainty in *R* simulations, especially in the far future.
We also compared these results with those evaluated based on Hamon PET (Fig. 7c),
and found some similar features. The differences in independent effects of *P* and *T*
between the two sets of results are mostly smaller than 5%, and both results show that
*T* effect would be twice as large as *P* effect at CONUS scale. This suggest that the bias
in PET model structure is not likely to turn over the relative importance of *P* and *T*
effects as long as $E_T$ model is properly calibrated. However, the projected decreases in
*R* (i.e., the 'Total' effects) are obviously more severe when using Hamon PET because
the positive effect of increasing humidity is not considered.
**3.3 Relative contributions of *P* and *T***
Table 2 summarizes the relative contributions of *P* and *T* to *R* change for the historical
and future periods in 18 WRRs and the entire CONUS. Historical changes in *P*, *T*, and
their effects on *R* were tested using PRISM climate data spanning from January 1960
to December 2010. Given the significant spatial and temporal variability in *R* trend
across the CONUS (Mauget, 2003;McCabe and Wolock, 2002, 2011;Gupta et al., 2015),
a consistent breakpoint is statistically unavailable. We hereby took 1985 as the
breakpoint year for all the watersheds and evaluated the multi-decadal mean changes





from 1961-1985 (pre-change period) to 1986-2010 (post-change period). Although the
selection of different breakpoints may cause certain deviations, the analysis can provide
a comparable benchmark for exploring the shifts in future scenarios at a multi-decadal
scale. Unsurprisingly, the results of these latest decades show the prevailing role of $P$
in nearly all the regions, with WRR14 being the only exception. In the future periods
(from baseline to S1~S4), however, results derived from both PM and Hamon PET
suggest that the role of $T$ rise will surpass $P$ and become the largest driver in most of
the regions (15~16 out of 18 WRRs) in the future. In contrast, a larger mean
contribution of $P$ can be occasionally found in the coastal regions (WRR1, 2, 18) and
the Southwest (WRR12, 15). Considering that the inconsistency among the different
GCMs may make the recognition of larger contributor dubious, we used Wilcoxon
signed-rank test (Gibbons and Chakraborti, 2011) to assess the statistical significance
of the difference between each pair of $P$ and $T$ contributions (i.e., 20 samples from the
20 GCMs). The test results reveal high agreement among GCMs on the prominent role
of $T$ across a major part of the CONUS, particularly the Midwest (WRR4~11) and the
Mountain West (WRR14,16) (underlined in Table 2).
At CONUS level, the mean contributions of $P$ and $T$ are projected to lie within
20%~24% and 43%~50% using PM PET, and 33%~40% and 55%~62% using Hamon
PET, suggesting a similar shift in the relative importance of these two key driving
factors. However, future changes in $Sh$, $Rs$, and $Ws$ account for another 16%~23%,
2%~7%, and 1%~4% of $R$ change respectively, and indirectly affect the attributions to
$P$ and $T$. For example, the $R$ increase in WRR1 would be completely attributed to $P$





increase if *Sh* was not considered, and thus lead to an overestimation of *P* contribution.
Also, we caution that spatially various levels of uncertainty are involved due to the
diverse changing directions and magnitudes of climatic variables projected by different
models.
**3.4 Spatial distribution of the major driving factors**
To further investigate the spatial pattern of future climatic controls on annual *R*, we
mapped the coverage of dominant driving factors and examined its consistency with
the changing trend in *R* at watershed scale (Fig. 8 & Table 3). Judging by multi-model
ensemble means, *P* and *T* are the largest driving factor in 10%~22% and 68%~89% of
the CONUS area. High consistency on their dominant roles (80% or more of the 20
GCMs agree on the sign) can be found in 4%~7% and 21%~41% of the CONUS,
respectivley. As *P* and *T* are projected to keep increasing, the coverages of *P*-dominant
and *T*-dominant areas are also expected to expand from 2030s to 2080s. A directional
change suggests that rising *T* will become more influential in the east (WRR1~6), while
*P* will prevail in more watersheds across the west (WRR13~18). Although the
aggregated effect of *Sh* is quite close to that of *P* at large scales, it is only expected to
play a dominant role in several watersheds (1% in area) across the borders between
WRR10 and WRR11 under RCP8.5/2080s.

19       The *P*-dominant areas that mainly distributed in the Southwest (WRR13,15) and

Pacific Coast (WRR17,18) show clear signals of increasing *R*, driven by the widespread
increase in *P*. One the other hand, only 61%~68% of the *T*-dominant areas coincide
with the areas of decreasing R, covering a large part of the Midwest (WRR7, 9~11) and



a number of watersheds scattered in the Mountain West (WRR14, 16, 17). Although $T$
is also identified as the most influential factor in the East (WRR1~5) by 2080s, the
combined effect of other four factors, primarily $P$ and $Sh$, is projected to exceed the $T$
effect and lead to an increase in $R$.
**4. Discussion**
**4.1 Spatial patterns of future changes in $R$**
This study characterizes and generalizes large-scale relationships among changing $P$, $T$,
and $R$ despite the large geographic differences. The coherence in the spatial dynamics
of $R$ trend and the corresponding climatic drivers shows a rough pattern: $T$ change
dominates $R$ decrease while $P$ and $Sh$ changes dominate $R$ increase. However, it should
be interpreted with limitations on time scale and underlying surface features. This
pattern does not hold true in all the watersheds due to the nonlinear complexity of $R$
response to climate change at various time scales, as well as the influence of other
watershed characteristics (e.g., topography, land-use, soil property). For example, slight
decreases in $P$ but somewhat increases in $R$ are projected in south Texas due to the
alteration of inner-annual climate variability. The role of $T$ may become more positive
in regions where water availability is dominated by snow melting (Barnett et al.,
2005;Lutz et al., 2014). Besides, local $R$ can be affected by other factors, such as land-
cover evolution and the direct effects of atmospheric composition on transpiration
(Gedney et al., 2006;Zhang et al., 2001;Zhang et al., 2015).
**4.2 The role of land cover and land use**
Land cover, LAI, and soil are important controls on catchment water balance and $R$





sensitivity to climate change (Zhang et al., 2001;Bosch and Hewlett, 1982;Cheng et al.,
2014). This study specifically focused on evaluating the separate and combined effects
of changing climates on $R$ within a static land cover/land use. We did not consider the
potential evolution of land cover and its interactions with water balance. We made no
explicit tabulation of the impact of land cover/land use on the $R$ responses to climate
change, but we did incorporate it as a key factor by estimating $E_T$ with a set of functions
of climate, LAI, and soil moisture capacity and deficit. Across the land cover classes,
the uncertainty ranges of independent contributions of $P$ (13%~30%) and $T$ (39%~51%)
are relatively small compared to the ranges across WRRs (18%~47% and 29%~52%).
This may be because the discrepancy across different land covers is largely offset by
the different climate backgrounds across the country. Evaluation of future land cover
change and its impact on $R$ is out of the scope of this study. However, our results imply
that the potential impact of land cover change might not be large enough to alter the
relative significance of $P$ and $T$ in controlling future continental water availability.
**4.3 Implications for water and land management**
Our results have important implications for water and land management across the
CONUS. Water resources planning may need to prepare different management
strategies for areas facing contrasting future hydrological conditions. Additional water
storage such as reservoirs and flood prevention measures may be needed in regions
expecting more $R$, while inter-basin water transfer, improving water use efficiency, and
other water conservation measures such as rain harvesting, and waste water recycling
should be implemented for areas expecting water shortages. The vast croplands across



central U.S. are likely to be threatened by rising $T$ and diminishing water availability

for irrigation and food production. Adaptations in cropping systems and irrigation

strategy are needed to secure food supply and increase resiliency to drought and

changing climate (Challinor et al., 2014;Teixeira et al., 2013). The drier and hotter

conditions may also result in increasing water stress, higher risks of tree insects and

disease outbreaks, and catastrophic wildfires in forests (Dale et al., 2001) (e.g., National

Forests in WRR14, 16, 17) and grasslands (e.g., in WRR10~11). Innovative land

management practices such as forest thinning and fuel management, irrigation, and

planting drought-tolerant species are vital to minimize the potential risk and

vulnerability to climate change and reduce the threats to ecosystems and society (Sun

et al., 2015a;Grant et al., 2013;Vose et al., 2016).

**4.4 Uncertainties and caveats**

Considerable uncertainty lies in the projection of future climate changes from the 20

GCMs. The uncertainty ranges under both RCP4.5 and RCP8.5 show significant

expansions over time from 2030s to 2080s. In particular, the large uncertainty in

predicting future $P$ may substantially compromise the reliability in evaluating either $R$

change or the roles of $P$ and $T$ (Karl and Riebsame, 1989;Piao et al., 2010). Although

the results allow us to draw some conclusions on the general patterns, uncertainties are

large and vary differently across space and time. There are certain limitations in this

evaluation that should be noted when interpreting the results. First, we did not

incorporate other sources of uncertainty, such as the methodology of downscaling

(Duan and Mei, 2014a;Chen et al., 2011), and structure and parameters of hydrologic



model (Jung et al., 2012). Although the selections of GCM and emission scenario are
more likely to be the largest sources of uncertainty in hydro-climatic modeling (Kay et
al., 2009;Wilby and Harris, 2006;Duan and Mei, 2014b), the other sources may also
affect the results to different extents. The roles of uncertainties from different sources
can be particularly equivocal when investigating seasonal/monthly variability and
extreme events (Bosshard et al., 2013;Giuntoli et al., 2015;Bae et al., 2011;Kay et al.,
2009). Second, we focused on the independent effects of potential climate changes in
this study, while assuming the inter-relationship among the meteorological variables
and water-balance components remains the same as in historical periods. In future
studies, improved climate datasets and better representation of the physical mechanisms
of climatic factors (e.g., radiation, Bohn et al., 2013; wind speed, McVicar et al., 2012)
are needed to reduce uncertainties.
**5. Conclusions**
This study evaluates the relative roles of precipitation and air temperature, as well as
solar radiation, wind speed, and humidity, in altering annual runoff across the CONUS
based on a large ensemble of simulations using data from both historical measurements
and CMIP5 GCMs projections. Despite the large uncertainty and spatial variability
involved, two robust conclusions can be drawn at the CONUS and regional scales on
multi-decadal basis. First, the role of temperature will outweigh that of precipitation in
a continued warming future in the 21$^{st}$ century, in spite that precipitation has been the
primary control of runoff variation during the latest decades. The projections from 20
climate models suggest a high degree of consistency on the increasing trends in both



precipitation and temperature, but the negative effect of temperature is expected to
exceed the positive effect of precipitation in most regions. Over the entire CONUS,
temperature is projected to be the largest contributor (43%~50%), followed by
precipitation (20%~24%), humidity (16%~23%), solar radiation (2%~7%), and wind
speed (1%~4%). Spatially, precipitation is likely to be the dominant driving factor for
runoff increase across the Pacific Coast and the Southwest, while temperature will be
more influential in the central CONUS and parts of the Mountain West. Particularly, the
vast areas of croplands and grasslands across the Midwest and forests in the Mountain
West might be under severe threat of water supply decline caused by warming.
Second, increasing humidity is expected to partially offset the additional evaporative
demand caused by warming, and consequently enhance runoff wide across the country.
Although the rising temperature is projected to be the largest control of runoff change
in the eastern CONUS, the combined effects of increasing humidity and precipitation
will surpass the detrimental effects of warming and result in a hydrologically wetter
future. This study also raises concern on the choice of PET method. It has been well
acknowledged in meteor-hydrology communities that temperature-based PET tends to
be oversensitive to temperature change. Our results further demonstrate that the main
risk of using temperature-based PET is overlooking the effects of other changing
climatic variables (mainly humidity in this case), which have not been as widely
measured as temperature and are relatively understudied, rather than overestimating the
effects of temperature.





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





# 1 Acknowledgements

This work was supported by the National Science Foundation EaSM program (AGS-
1049200) awarded to North Carolina State University, and the Eastern Forest
Environmental Threat Assessment Center (EFETAC), USDA Forest Service, Raleigh,
NC. The MACAv2-LIVNEH dataset was produced under the Northwest Climate
Science Center (NW CSC) US Geological Survey Grant Number G12AC20495. Partial
support was provided by the Natural Science Foundation of Jiangsu Province, China
(BK20151525); the Pine Integrated Network: Education, Mitigation, and Adaptation
project (PINEMAP), which is a Coordinated Agricultural Project funded by the USDA
National Institute of Food and Agriculture, Award #2011-68002-30185. The authors
would like to give special thanks to Dr. Dennis Lettenmaier for his valuable comments
and suggestions.




**Tables**
**Table 1.** List of the 20 climate models and the changes in mean annual precipitation and temperature
over the conterminous United States (CONUS) from the baseline scenario (B) to future scenarios S1
(RCP4.5/2030s), S2 (RCP4.5/2080s), S3 (RCP8.5/2030s), and S4 (RCP8.5/2080s).

| GCM | Country | Precipitation (mm yr⁻¹) | | | | | Temperature (°C) | | | | |
|---|---|---|---|---|---|---|---|---|---|---|---|
| | | B | S1 | S2 | S3 | S4 | B | S1 | S2 | S3 | S4 |
| bcc-csm1-1 | China | 787 | -3 | +13 | +33 | -5 | 11.4 | +1.7 | +2.4 | +1.9 | +4.8 |
| bcc-csm1-1-m | China | 786 | +18 | -18 | +29 | +33 | 11.4 | +1.5 | +2.4 | +1.7 | +4.3 |
| BNU-ESM | China | 798 | +51 | +42 | +25 | +45 | 11.5 | +1.9 | +3.2 | +2.0 | +5.4 |
| CanESM2 | Canada | 800 | +14 | +42 | +19 | +83 | 11.3 | +2.3 | +3.5 | +2.4 | +5.8 |
| CCSM4 | USA | 783 | +29 | +29 | +18 | +58 | 11.5 | +1.5 | +2.5 | +1.9 | +4.6 |
| CNRM-CM5 | France | 780 | +46 | +56 | +40 | +85 | 11.4 | +1.4 | +2.8 | +1.6 | +4.6 |
| CSIRO-Mk3-6-0 | Australia | 780 | +14 | +84 | +24 | +74 | 11.2 | +2.0 | +3.4 | +2.0 | +5.6 |
| GFDL-ESM2M | USA | 787 | +6 | +20 | +32 | +31 | 11.3 | +1.6 | +2.2 | +1.7 | +4.2 |
| GFDL-ESM2G | USA | 791 | +21 | +36 | +38 | +12 | 11.4 | +1.2 | +1.7 | +1.2 | +3.7 |
| HadGEM2-ES | UK | 784 | +16 | +7 | +18 | +7 | 11.3 | +2.2 | +3.8 | +2.5 | +6.8 |
| HadGEM2-CC | UK | 779 | +23 | +39 | +5 | +32 | 11.3 | +2.3 | +4.2 | +2.7 | +6.7 |
| inmcm4 | Russia | 779 | -7 | +4 | +0 | +13 | 11.4 | +0.9 | +1.7 | +1.1 | +3.4 |
| IPSL-CM5A-LR | France | 780 | +8 | +14 | +13 | -8 | 11.5 | +1.8 | +3.0 | +1.8 | +5.8 |
| IPSL-CM5A-MR | France | 789 | -4 | +13 | -25 | -70 | 11.3 | +1.9 | +3.2 | +2.3 | +6.0 |
| IPSL-CM5B-LR | France | 781 | +23 | +62 | +34 | +82 | 11.4 | +1.5 | +2.4 | +1.7 | +4.4 |
| MIROC5 | Japan | 788 | +9 | +10 | +24 | +6 | 11.2 | +2.3 | +3.6 | +2.4 | +5.7 |
| MIROC-ESM | Japan | 791 | +56 | +37 | +30 | +9 | 11.3 | +2.1 | +4.1 | +2.6 | +6.6 |
| MIROC-ESM-CHEM | Japan | 784 | +12 | +38 | +26 | +10 | 11.4 | +2.4 | +4.0 | +2.7 | +6.9 |
| MRI-CGCM3 | Japan | 783 | +20 | +47 | +38 | +87 | 11.4 | +0.8 | +1.7 | +1.0 | +3.2 |
| NorESM1-M | Norway | 784 | +13 | +31 | +25 | +63 | 11.3 | +1.8 | +3.1 | +2.2 | +5.1 |





**Table 2.** Comparison of multi-model averaged contributions (%) of precipitation ($P$) and temperature ($T$) to changes in runoff in the 18 Water Resource Regions (WRRs) and entire CONUS in historical period (1961-2010) and future scenarios S1 (RCP4.5/2030s), S2 (RCP4.5/2080s), S3 (RCP8.5/2030s), and S4 (RCP8.5/2080s). The larger contributor identified by multi-model ensemble means is bolded, and the significant larger contributor identified by Wilcoxon signed-rank test (at 5% significance) is underlined.

Projections based on PM PET

| WRR | Historical P | Historical T | S1 P | S1 T | S2 P | S2 T | S3 P | S3 T | S4 P | S4 T |
|---|---|---|---|---|---|---|---|---|---|---|
| 1 | 88 | 9 | 36 | 36 | 36 | 38 | 34 | 38 | 31 | 42 |
| 2 | 80 | 17 | 27 | 40 | 28 | 41 | 30 | 39 | 28 | 43 |
| 3 | 60 | 30 | 31 | 37 | 26 | 41 | 30 | 38 | 24 | 44 |
| 4 | 83 | 13 | 24 | 44 | 23 | 46 | 29 | 41 | 23 | 47 |
| 5 | 73 | 22 | 23 | 42 | 23 | 44 | 29 | 40 | 25 | 46 |
| 6 | 64 | 30 | 28 | 40 | 27 | 42 | 32 | 38 | 26 | 45 |
| 7 | 89 | 6 | 23 | 47 | 19 | 51 | 23 | 48 | 20 | 52 |
| 8 | 48 | 37 | 27 | 39 | 23 | 43 | 24 | 42 | 24 | 46 |
| 9 | 89 | 8 | 22 | 47 | 20 | 49 | 26 | 45 | 20 | 43 |
| 10 | 81 | 6 | 19 | 47 | 18 | 50 | 18 | 49 | 20 | 46 |
| 11 | 88 | 4 | 20 | 42 | 19 | 45 | 18 | 44 | 18 | 47 |
| 12 | 74 | 14 | 35 | 29 | 27 | 35 | 30 | 32 | 27 | 39 |
| 13 | 71 | 18 | 25 | 36 | 27 | 38 | 26 | 35 | 22 | 42 |
| 14 | 30 | 51 | 21 | 48 | 25 | 48 | 20 | 49 | 24 | 49 |
| 15 | 72 | 17 | 28 | 33 | 36 | 32 | 33 | 32 | 36 | 29 |
| 16 | 65 | 23 | 21 | 45 | 24 | 46 | 23 | 45 | 29 | 43 |
| 17 | 91 | 7 | 28 | 42 | 28 | 43 | 29 | 42 | 31 | 42 |
| 18 | 95 | 4 | 47 | 29 | 43 | 32 | 46 | 30 | 46 | 30 |

Projections based on Hamon PET

| WRR | S1 P | S1 T | S2 P | S2 T | S3 P | S3 T | S4 P | S4 T |
|---|---|---|---|---|---|---|---|---|
| 1 | 61 | 38 | 58 | 40 | 57 | 41 | 53 | 46 |
| 2 | 47 | 50 | 49 | 50 | 51 | 47 | 46 | 52 |
| 3 | 43 | 49 | 38 | 56 | 41 | 52 | 32 | 60 |
| 4 | 44 | 54 | 42 | 57 | 50 | 48 | 40 | 58 |
| 5 | 40 | 57 | 38 | 59 | 46 | 51 | 37 | 60 |
| 6 | 41 | 54 | 40 | 56 | 46 | 49 | 37 | 58 |
| 7 | 40 | 57 | 32 | 65 | 37 | 59 | 32 | 65 |
| 8 | 38 | 53 | 34 | 58 | 35 | 56 | 29 | 61 |
| 9 | 37 | 56 | 34 | 61 | 40 | 53 | 33 | 57 |
| 10 | 35 | 57 | 32 | 62 | 32 | 60 | 33 | 59 |
| 11 | 30 | 55 | 29 | 60 | 27 | 58 | 26 | 63 |
| 12 | 44 | 38 | 37 | 46 | 38 | 42 | 31 | 51 |
| 13 | 35 | 53 | 36 | 56 | 37 | 53 | 28 | 61 |
| 14 | 31 | 64 | 36 | 60 | 32 | 64 | 31 | 61 |
| 15 | 35 | 52 | 41 | 48 | 43 | 47 | 37 | 49 |
| 16 | 34 | 59 | 36 | 58 | 32 | 60 | 38 | 51 |
| 17 | 44 | 54 | 44 | 54 | 45 | 53 | 47 | 51 |
| 18 | 58 | 36 | 54 | 41 | 56 | 39 | 54 | 42 |



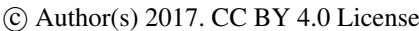





**Table 3.** Cross comparison of the areal proportions (%) with different dominant driving factors and
changing directions of runoff (*R*) in the future scenarios S1 (RCP4.5/2030s), S2 (RCP4.5/2080s), S3
(RCP8.5/2030s), and S4 (RCP8.5/2080s). The climate variable is identified as 'dominant' when 80% or
more of the 20 GCMs agree that it is the largest driving factor of runoff change. The areas where a
variable is the largest driving factor by ensemble mean is marked in the brackets, and the areas with a
significant dominant factor is bolded.

| Scenario | S1 | S2 | S3 | S4 |
|---|---|---|---|---|
| *Precipitation* | | | | |
| $R\nearrow$ [a] | **4** (10) | **7** (17) | **6** (15) | **6** (21) |
| $R\searrow$ | **0.2** (0.2) | 0 | **0.2** (0.2) | 0 (0.7) |
| *Temperature* | | | | |
| $R\nearrow$ | **9** (51) | **15** (45) | **7** (55) | **13** (26) |
| $R\searrow$ | **15** (38) | **23** (37) | **14** (30) | **28** (42) |
| *Solar radiation* | | | | |
| $R\nearrow$ | 0 | 0 | 0 | 0 |
| $R\searrow$ | 0 | 0 | 0 | 0 |
| *Wind speed* | | | | |
| $R\nearrow$ | 0 | 0 | 0 | 0 |
| $R\searrow$ | 0 | 0 | 0 | 0 |
| *Specific humidity* | | | | |
| $R\nearrow$ | 0 (0.2) | 0 (2) | 0 (0.2) | **0.8** (5) |
| $R\searrow$ | 0 (0.2) | 0 (0.4) | 0 | **1** (5) |

[a] "$\nearrow$" and "$\searrow$" indicate increase and decrease in the multi-model means of runoff, respectively.





**Figures**

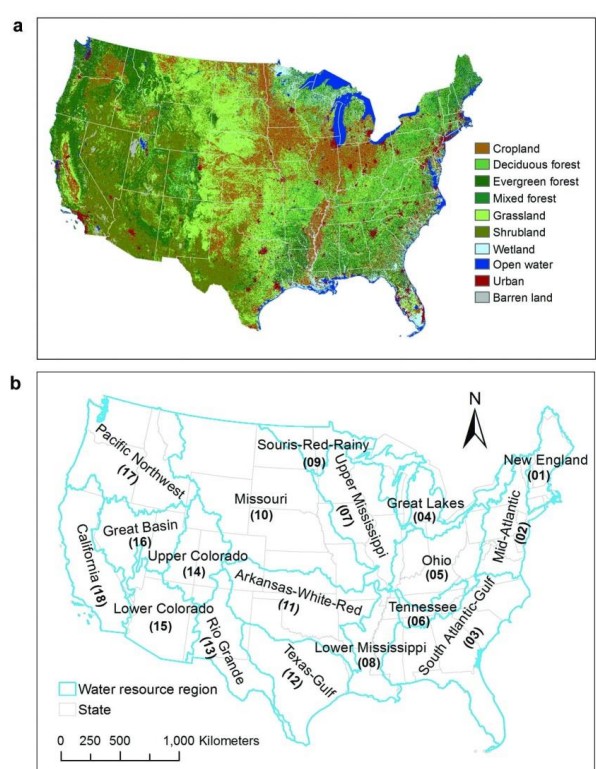

**Figure 1.** (**a**) Land-cover distribution in the conterminous United States (CONUS) from the 2006
National Land Cover Database (http://www.mrlc.gov/nlcd06_data.php), and (**b**) location of the 18 Water
Resource Regions (WRRs).





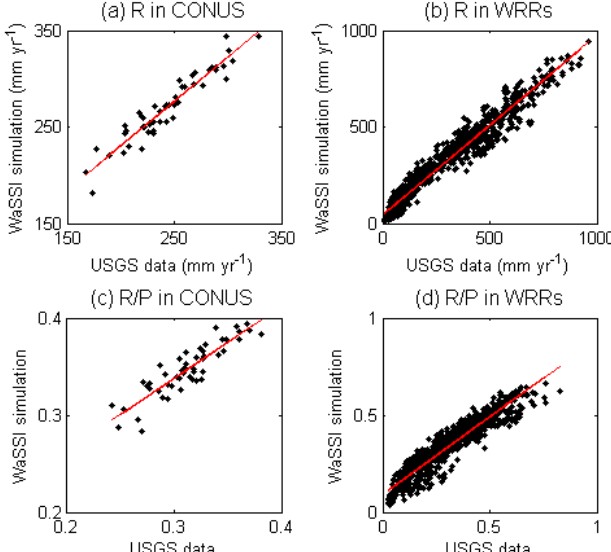

**Figure 2.** Validations of the WaSSI model at the conterminous United States (CONUS) and Water Resource Region (WRR) levels. **a-b**, Comparisons of simulated annual runoff ($R$) (mm yr$^{-1}$) against USGS observed data in 1961-2010 over the entire CONUS (**a**) and in 18 WRRs (**b**). **c-d**, Comparisons of simulated runoff coefficient (runoff/precipitation, R/P) against that derived from USGS observed data in the CONUS (**c**) and WRRs (**d**).



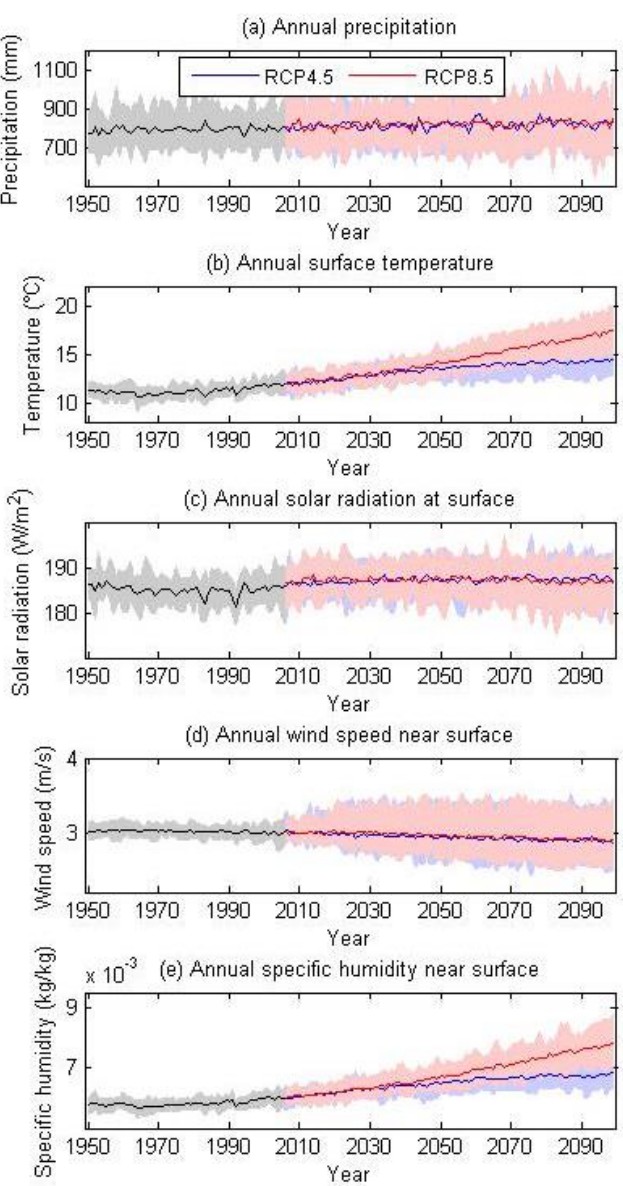

**Figure 3.** Temporal variations of annual mean precipitation (**a**), surface air temperature (**b**), solar
radiation at surface (**c**), wind speed near surface (**d**), and specific humidity near surface (**e**) over the
CONUS. Thick lines and the shading denote the multi-model ensemble means and uncertainty ranges of
the 20 GCMs, respectively.



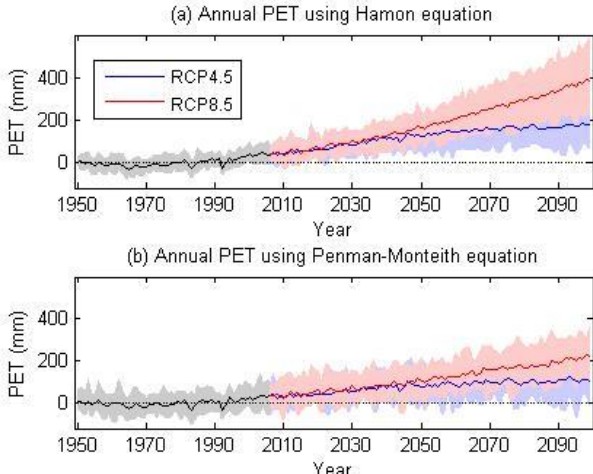

**Figure 4.** Temporal variations of changes in annual potential evapotranspiration (PET) over the CONUS

against the baseline level (1970-1999). Thick lines and the shading denote the ensemble means and

uncertainty ranges of the 20 GCMs, respectively.



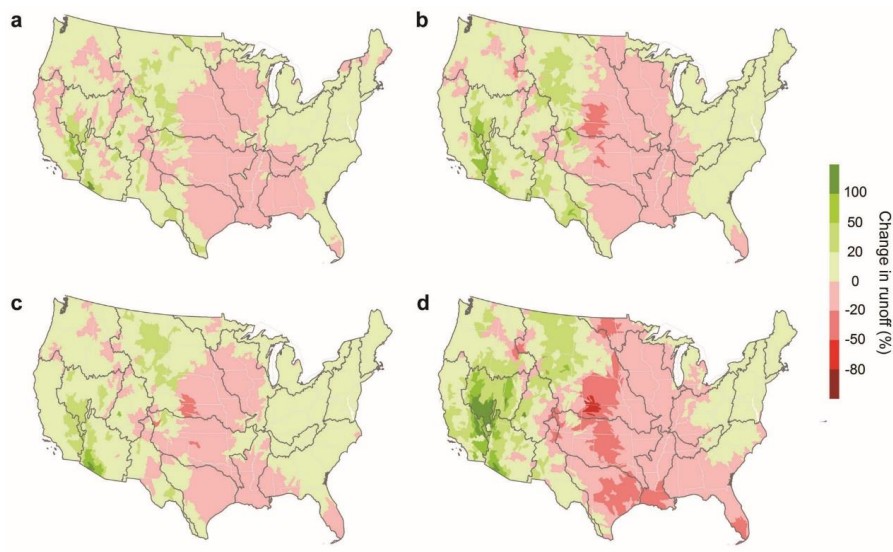

**Figure 5.** Projected changes in multi-year mean annual runoff (%) at HUC-8 watershed scale. **a-d**,
Changes from the baseline to S1 (RCP4.5/2030s) (**a**), S2 (RCP4.5/2080s) (**b**), S3 (RCP8.5/2030s) (**c**),
and S4 (RCP8.5/2080s) (**d**) scenarios. The maps display the multi-model mean changes from the 20
GCMs.



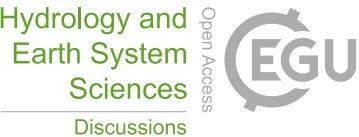

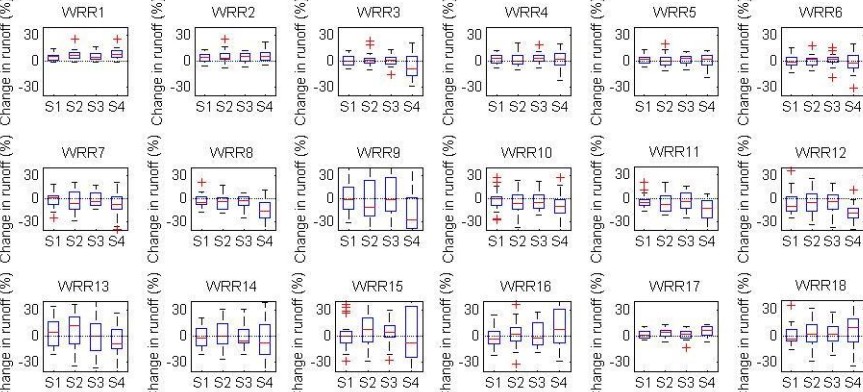

**Figure 6.** Area-averaged changes in runoff in the 18 Water Resource Regions (WRRs) in the future
scenarios. The four future scenarios are denoted by S1 (RCP4.5/2030s), S2 (RCP4.5/2080s), S3
(RCP8.5/2030s), and S4 (RCP8.5/2080s) in the x-axis. The vertical spread of the box-whisker plots
shows the different results projected from the 20 GCMs, with the boxes covering the ranges from 25%
quartile to 75% quartile of the distributions (Inter-Quartile Range, IQR) and the red lines within each
box marking the median values. Points outside the whiskers are taken as extreme outliers and marked by
plus signs.



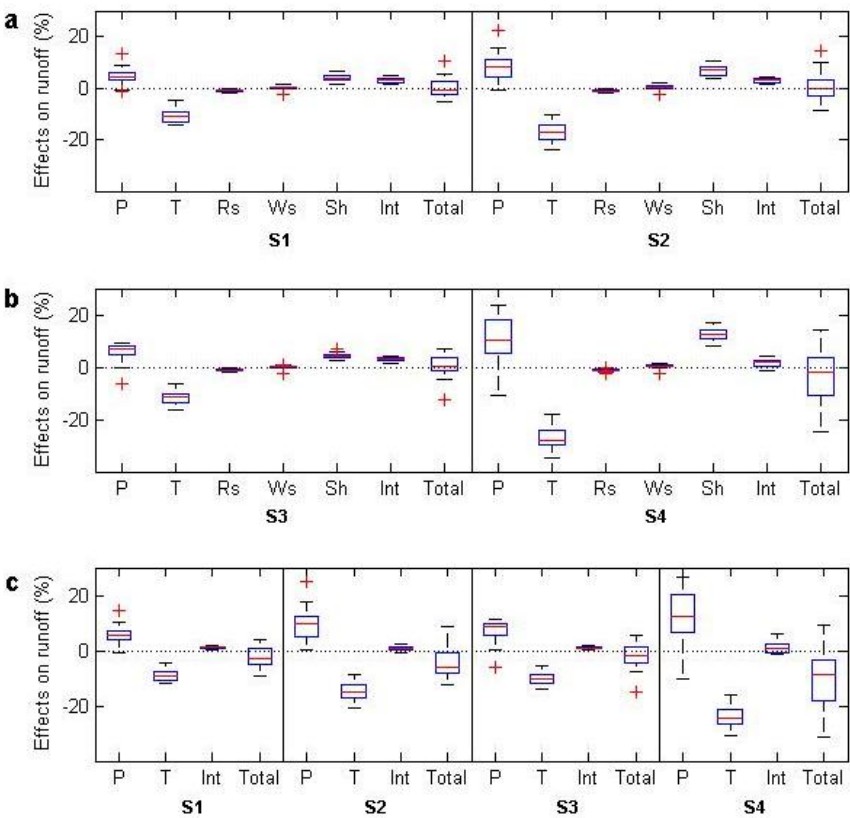

**Figure 7.** Independent effects of the climate variables over the conterminous United States (CONUS) in the future scenarios S1 (RCP4.5/2030s), S2 (RCP4.5/2080s), S3 (RCP8.5/2030s), and S4 (RCP8.5/2080s). **a-b**, Effects of precipitation (*P*), temperature (*T*), solar radiation (*Rs*), wind speed (*Ws*), specific humidity (*Sh*), interactions among the variables (*Int*), and their sum (*Total*) on runoff based on the projections of Penman-Monteith PET. **c**, Effects of precipitation (*P*), temperature (*T*), interaction between *P* and *T* (*Int*), and their sum (Total) on runoff based on the projections of Hamon PET.



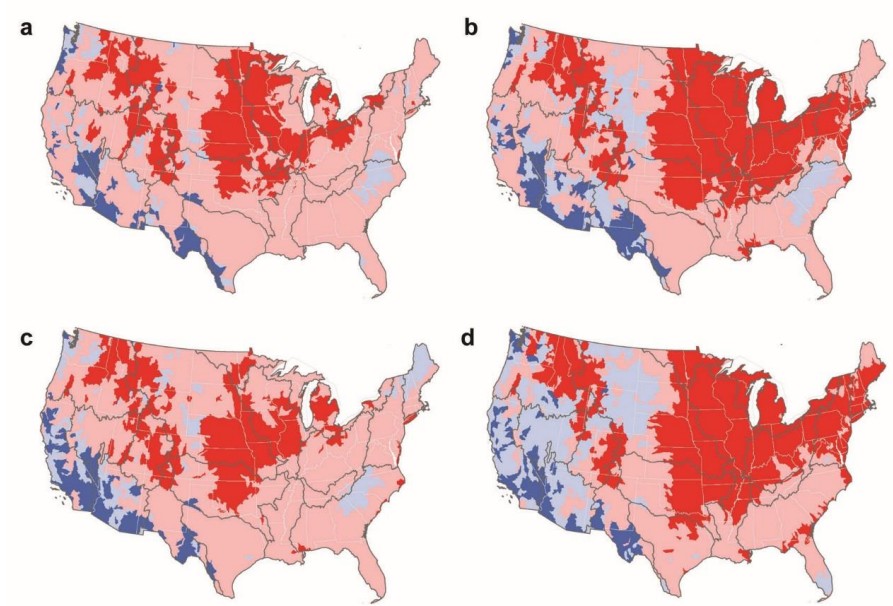

**Figure 8.** Relative importance of *P* and *T* in affecting runoff change across the HUC-8 watersheds in the

future scenarios of S1 (RCP4.5/2030s) (**a**), S2 (RCP4.5/2080s) (**b**), S3 (RCP8.5/2030s) (**c**), and S4

(RCP8.5/2080s) (**d**). The watersheds under larger influence of *P* and *T* are marked with blue and red

colors, respectively. The dark colors denote the areas where 80% or more of the 20 GCMs agree on the

sign, while the light colors denote the results of ensemble average.

