# Peer review of "Future shift of the relative roles of precipitation and temperature in"

_Hydrology and Earth System Sciences, 2017_

## Referee Comment (RC1) · Anonymous Referee #1 · 10 Aug 2017

Review of the manuscript "Future shift of the relative roles of precipitation and temperature in controlling annual runoff in the conterminous United States" by Duan et al.

In this manuscript, Duan et al. evaluated the relative importance of climate variables (precipitation, temperature, humidity, wind speed and solar radiation) in changing the annual runoff volume under future climate change scenarios in the United States. They apply an ecohydrological model on a monthly basis and additionally run the model with two different potential evaporation inputs. Temperature will outweigh the historic importance of precipitation for runoff variability in the future in most of the U.S. although

increased humidity can partly reduce evaporative demand and therefore lead to an increase in runoff. The way potential evaporation is calculated has an effect on runoff simulations, but using a variety of climate variables is considered as a more important factor in climate studies.

This is an interesting study and I like the clear and well described concept. The results are illustrated and described in detail for different levels of spatial aggregations and clearly support the conclusions. The main limitations as well as the implications of the study are well discussed. To further improve the manuscript I have some suggestions listed below including the combination of some of the tables and figures to condense the information and evaluating the results based on hydroclimatic regions.

I hope that the comments below will be helpful for the authors to improve their manuscript.

**Major comments:**

P2 Abstract: I think the abstract would benefit from some more precise or detailed information. E.g. L3: name of the model and simulated time resolution (month); L13-16: It is stated that precipitation will lead to an increase in runoff, while for temperature it only states that there is a large effect but does not explicitly say in which direction (increase or decrease). L18-19: Why do the Midwest and South-Central regions have a severe runoff depletion?

P5 L16-18: This comment is about the WaSSi model: a) What are the benefits of using the WaSSI model and not a simpler model (e.g. a model with less input data and probably less parameters) in your study? b) Snow routine and the ET model are well described. Could you also give some more information about the structure of the SAC-SMA model? A schematic of the WaSSi model could help the reader to more easily understand the model structure, its disaggregation into land cover types,

number of parameters, etc. c) Does each catchment (HUC 8 level) have only one single parameter set which is used for runoff simulations and why? Using many more parameter sets could make the results more robust and reliable.

P5 L16-18: Could you maybe say a few words about the selection criteria of your study catchments? E.g. did you use all catchments available in the U.S. at HUC 8 level and is human influence on the study catchments problematic for your results? Please also provide the source of your catchment data (catchment outline, runoff data, etc.) in the text and the references.

P8 L12-14: EInt is calculated as the difference between the change in runoff and the combined effect of the climate variables. Doesn't this assume a perfect model, i.e. that the selected variables can explain all the runoff changes?

P9 L4-6: This sentence lists minimum and maximum temperature as climate variables, but Fig. 3 refers to surface air temperature only. Does Fig. 3 show the mean of the minimum and the maximum temperature? Which temperature was used for runoff simulations?

P14 L20-22; P15 L9-11; P15 L19-22: I strongly recommend to make more explicit references to tables and figures to clearly indicate the reader where to find the described information. E.g. P14 L20-22: Table 2 does not provide information about Sh, Rs and Ws. P15 L9-11 and P15 L19-22: I couldn't find the indicated percentages in Fig. 8 or Table 3.

P19-P20 Conclusions: Similar to the abstract I would recommend to be more precise. E.g. L17: to what exactly do the large uncertainty and spatial variability refer to? (projected changes in runoff?). L1: what is negatively affected by the increasing temperature? (annual runoff?). L6-7: temperature will decrease runoff. L17: temperature based PET tends to be oversensitive to changes in temperature compared to Penman-Monteith.

P27 Table 1: Table 1 and Fig. 3 contain to a large degree redundant information. To me it is most important to have an impression of the general trends of the 5 climate variables T, P, Sh, Rs and Ws in the two RCPs while the origin country of a GCM is not relevant for the interpretation of the results. Since Fig. 3 provides the trend information of the climate variables, I recommend to delete Table 1 and list the names of the GCMs in the text of section 2.3.

P31 Fig. 1: I am not sure if these two figures are necessary. Fig. 1a is only used in the context of the WaSSi model, where the individual land cover types are listed. Since the map is not further used in the results or discussion part I probably would remove it. Fig. 1b could maybe also be skipped - WRR names could be added to Table 2 and WRR IDs could be added to the maps of Fig.5 and Fig.8. Having the IDs directly in the maps would support the readability of the results where usually a reference to the WRR is made.

P36 Fig. 6: This is a more general comment on the use of WRRs and therefore also applies to Table 2 and the corresponding results parts. I wonder how much the averaged results on the level of WRR actually tell us? WRR can be considered as very large watersheds spanning a wide range of land cover types and hydroclimates. The runoff response of subbasins of a WRR to changes in climate variables can therefore be very diverse, which can be seen in Fig. 8. From a hydrological perspective it would be interesting to see exactly these relationships between changes in runoff response and hydroclimate, land cover, etc. Averaging the runoff response over a WRR makes conclusions about possible relationships difficult. In my opinion it would be worth to analyze the runoff response to changes in P and T in dependence of the hydroclimate (e.g. see studies of Coopersmith et al., 2014; Sawicz et al., 2014) or the Köppen Geiger climate zones.

P38 Fig. 8: The information of Table 3 and Fig. 8 is very similar. Is it possible to combine the two? The fact that solar radiation, wind speed and specific humidity have little effect on changes in runoff response is already illustrated in Fig. 7 and therefore

does not need to be repeated in Table 3. The areal proportions for precipitation and temperature as driving factors could be directly added to the maps in Fig. 8.

**Minor comments:**

P8 L3-16: The terms "climate variables" and "driving factors" are used interchangeably as synonyms, which can be confusing. I recommend to use only one of the two terms.

P8 L7-8: I recommend to write "...independent effects E of each driving factor Ci..."

P8 L15: Based on equation 3 I assume that the contributions of the climate variables are quantified by the absolute relative weights.

P9 L13-19: The first two sentences about sensitivity are to my perception not so relevant and could be deleted. I don't fully understand the last sentence - does pooling mean averaging of results?

P11 L2: I would not use abbreviations in the title.

P20 L7-9: I think it is not necessary to mention in the conclusion that the Midwest has vast areas of croplands and grasslands, because this was not a major finding of the study.

P32 Fig. 2: The R-square values mentioned at P7 L16 could be added to the graph.

P36 Fig. 6: The figure caption explains the elements of a boxplot. If you think this is needed you should also add the explanation in Fig. 7 to be consistent. Additionally, I recommend to use the same y-axis labels in the two figures.

Please use the HESS guidelines for all abbreviations and units. E.g. P33 Fig. 3: adapt units from W/m2 to W m-2.

According to the HESS guidelines, authors are encouraged to briefly describe the

contribution of each co-author in a section called "author contributions".

**References:**

Coopersmith, E. J., B. S. Minsker, and M. Sivapalan. "Patterns of regional hydroclimatic shifts: An analysis of changing hydrologic regimes." Water Resources Research, 50.3, 1960-1983, 2014.

Sawicz, K. A., Kelleher, C., Wagener, T., Troch, P., Sivapalan, M., Carrillo, G.: Characterizing hydrologic change through catchment classification. Hydrology and Earth System Sciences, 18(1), 273-285, 2014

---

## Referee Comment (RC2) · B.L. Finlayson (Referee) · 20 Aug 2017

The overall academic content of this paper is sound, exploring the possible future runoff across the coterminous United States under conditions that may develop as predicted global climate changes unfold. However, my concerns with this paper relate to the way this material is presented. The authors appear to be unaware that they are writing to a global audience, and not to a group who, like themselves, are very familiar with the geography of the coterminous USA and with the systems used for identifying watersheds and location in the USA. I list below a series of points to illustrate my concerns.

P 1 Lines 5-6: The use of the phrase "hydrologic paradigms" seems inappropriate here.

[Figure]

What is at issue here is the strength or intensity of different hydrological processes. Paradigms are something rather different.

P 1 Line 7: "intensification of hydrologic cycle". What does this phrase mean?

P 1 Line 12: The use of "sustainably" in this context seems rather out of place. There are a lot of surface water sources and shallow aquifers that are being used very unsustainably.

P 4 Lines 16-17 "the rate of decadal change of temperature over the CONUS has reached $-0.03 \sim +0.28$ °C since 1960s". I'm not sure what this means, it needs to be more clearly stated.

The authors assume that the readers have an intimate knowledge of some of the materials they are working with. So, for example, they use the term "8-digit Hydrologic Unit Code (HUC-8) watersheds" and "2-digit HUC Watershed". I have no idea what these are and I suspect I'm not the only one. The paper needs to be written for an international audience and not a just a group of those specialising in North American hydrology.

P 8 I do not follow the discussion from Line 3 to Line 17. Especially this term (Line 12) - $R(C_1 t_1, \ldots, C_i t_2, \ldots, C_N t_1) - R(C_1 t_1, \ldots, C_i t_1, \ldots, C_N t_1)$. What is going on here needs to be explained more clearly, or is there a misprint?

P 8 Line 20 "statistically downscaled" What does this mean? Is this a way of saying that the means or the medians were used?

P 9 lines 1-2 "RCP4.5 and RCP8.5 were adopted as representatives of the intermediate and high emission scenarios respectively". At this point in the paper the readers have no idea what RCP4.5 and RCP8.5 are. There is some explanation later in the paragraph but it is not particularly clear. These terms need to be defined before they are used.

Similarly, in Section 3, where the results are presented, Water Resource Regions

(WRR) are referred to by their numbers and sometimes also the name of a general region, such as Midwest, Mountain West or coastal regions, in this case with no indication which bits of the coastal US are being referred to.

The writing style is rather unsatisfactory with frequent lack of the definite article and missing and incorrect words. Here is an example: "For example, slight decreases in P but somewhat increases in R are projected in south Texas due to the alteration of inner-annual climate variability." I suspect that this, and the many similar cases in the text, come about from reviewing the text using the word processor's spelling check rather than careful reading by the authors.

In Section 4.3 the authors argue that the results presented here indicate that "Additional water storage such as reservoirs and flood prevention measures may be needed in regions expecting more R". That may be the case but there is no evidence in this study that relates to flood behaviour and simply an increase in runoff does not say anything one way or the other about how floods will behave.

---

## Author Comment (AC1) · 28 Aug 2017

We are grateful to the reviewer for the thoughtful and detailed comments. Here we present our response to the comments and plan of revision.

**Reviewer comment**
*In this manuscript, Duan et al. evaluated the relative importance of climate variables (precipitation, temperature, humidity, wind speed and solar radiation) in changing the annual runoff volume under future climate change scenarios in the United States. They apply an ecohydrological model on a monthly basis and additionally run the model with two different potential evaporation inputs. Temperature will outweight the historic importance of precipitation for runoff variability in the future in most of the U.S. although increased humidity can partly reduce evaporative demand and therefore lead to an increase in runoff. The way potential evaporation is calculated has an effect on runoff simulations, but using a variety of climate variables is considered as a more important factor in climate studies.*

*This is an interesting study and I like the clear and well described concept. The results are illustrated and described in detail for different levels of spatial aggregations and clearly support the conclusions. The main limitations as well as the implications of the study are well discussed. To further improve the manuscript, I have some suggestions listed below including the combination of some of the tables and figures to condense the information and evaluating the results based on hydroclimatic regions. I hope that the comments below will be helpful for the authors to improve their manuscript.*

*P2 Abstract: I think the abstract would benefit from some more precise or detailed information. E.g. L3: name of the model and simulated time resolution (month); L13-16: It is stated that precipitation will lead to an increase in runoff, while for temperature it only states that there is a large effect but does not explicitly say in which direction (increase or decrease). L18-19: Why do the Midwest and South-Central regions have a severe runoff depletion?*
**Author reply**
We will clarify these issues in the revised abstract.

**Reviewer comment**
*P5 L16-18: This comment is about the WaSSi model: a) What are the benefits of using the WaSSI model and not a simpler model (e.g. a model with less input data and probably less parameters) in your study? b) Snow routine and the ET model are well described. Could you also give some more information about the structure of the SAC-SMA model? A schematic of the WaSSi model could help the reader to more easily understand the model structure, its disaggregation into land cover types, number of parameters, etc. c) Does each catchment (HUC 8 level) have only one single parameter set which is used for runoff simulations and why? Using many more parameter sets could make the results more robust and reliable.*
**Author reply**
We will add more information about the SAC-SMA model. WaSSI was integrated from a snow model, a ET model, and SAC-SMA model. It was specifically developed to capture large-scale water balance in the conterminous US based on empirical and physically based parameters. The

parameters of snow model were calibrated for different HUC-2 regions; the parameters of ET model were calibrated for different HUC-2 regions and land-cover types; the parameters of SAC-SMA model were derived from soil properties at 1×1 km resolution (State Soil Geographic Database, STATSGO). Comparing with other simpler models, such as widely used Budyko equations, users can directly use these tested parameters available for the conterminous US and do not need to go through the processes of parameterization for each single watershed. We agree that uncertainty exists in these parameters, but the accuracy has been proved satisfactory.

Please see the second paragraph of Section 2.1 for a more detailed description of the parameters. Readers are referred to McCabe and Markstrom (2007) for the snow model, Sun et al. (2011) for the ET model, Caldwell et al. (2012) for the structure of WaSSI, and Koren et al. (2003) for the SAC-SMA parameters.

References:

*Caldwell, P., Sun, G., McNulty, S., Cohen, E., and Moore Myers, J.: Impacts of impervious cover, water withdrawals, and climate change on river flows in the conterminous US, Hydrology and Earth System Sciences, 16, 2839-2857, 2012.*
*Koren, V., Smith, M., and Duan, Q.: Use of a priori parameter estimates in the derivation of spatially consistent parameter sets of rainfall-runoff models, Calibration of Watershed Models, 6, AGU, Washington, D.C., 239-254 pp., 2003.*
*McCabe, G. J., and Markstrom, S. L.: A monthly water-balance model driven by a graphical user interface, Geological Survey (US)2331-1258, 2007.*
*Sun, G., Caldwell, P., Noormets, A., McNulty, S. G., Cohen, E., Moore Myers, J., Domec, J. C., Treasure, E., Mu, Q., and Xiao, J.: Upscaling key ecosystem functions across the conterminous United States by a water-centric ecosystem model, Journal of Geophysical Research, 116, 10.1029/2010JG001573, 2011.*

**Reviewer comment**

*P5 L16-18: Could you maybe say a few words about the selection criteria of your study catchments? E.g. did you use all catchments available in the U.S. at HUC 8 level and is human influence on the study catchments problematic for your results? Please also provide the source of your catchment data (catchment outline, runoff data, etc.) in the text and the references.*

**Author reply**

Yes, we calculated all the 2,099 HUC-8 watersheds in the conterminous US. We will add a section "2.1.1 Study area" to briefly introduce the American hydrologic unit system to international readers.

The hydrological model was validated by discharge in 10 representative watersheds without significant human influence (Caldwell et al., 2012) to make sure that the model reproduces precipitation-runoff processes. In this study, we focused on the potential changes in climatic variables, and did not consider the potential effects of land cover change (e.g., urbanization, forestation) and other human activities (e.g., river regulation and water transfer projects).

**Reviewer comment**

*P8 L12-14: EInt is calculated as the difference between the change in runoff and the combined effect of the climate variables. Doesn't this assume a perfect model, i.e. that the selected variables can explain all the runoff changes?*

**Author reply**

This study was limited to the effects of these changing climatic variables with the assumption that other factors (i.e., soil properties, land cover and land use) remain constant. Thus, the runoff

change we presented as 'total' effect was a result of the changing climate only, and this change in runoff was attributed to different climatic variables to compare their relative roles. Runoff response to these changes depends on model structure (mainly sensitivity of PET) and the magnitudes of these changes.

**Reviewer comment**
*P9 L4-6: This sentence lists minimum and maximum temperature as climate variables, but Fig. 3 refers to surface air temperature only. Does Fig. 3 show the mean of the minimum and the maximum temperature? Which temperature was used for runoff simulations?*
**Author reply**
We only showed the general trends of annual mean values in Fig. 3. Monthly averages of daily minimum, maximum, and mean (estimated as the average of min + max) temperature were all used in runoff simulations, and the '*T* effect' represented the effect of changes in all of them. We will clarify this at the end of Section 2.3.2.

**Reviewer comment**
*P14 L20-22; P15 L9-11; P15 L19-22: I strongly recommend to make more explicit references to tables and figures to clearly indicate the reader where to find the described information. E.g. P14 L20-22: Table 2 does not provide information about Sh, Rs and Ws. P15 L9-11 and P15 L19-22: I couldn't find the indicated percentages in Fig. 8 or Table 3.*
**Author reply**
We will rephrase the caption of Table 3 and the references in the manuscript to make it clearer.
  In P14, the contributions of Sh, Rs and Ws were derived from their independent effects. We did not add another figure to show the detailed contributions of each climatic variable because their relative contributions are consistent with the magnitudes of independent effects shown in Fig. 4.
  In P15, all the areal percentages can be found in Table 3. Note that we marked two kinds of areal proportions in the table to keep it concise. The areas where a variable is the largest driving factor identified by multi-model averages is marked in the brackets (L9). The areas where a variable is identified as the 'dominant' factor are bolded (L11). A climate variable is identified as the 'dominant' one only when 80% or more of the 20 GCMs agree that it will be the largest driving factor of runoff change.

**Reviewer comment**
*P19-P20 Conclusions: Similar to the abstract I would recommend to be more precise. E.g. L17: to what exactly do the large uncertainty and spatial variability refer to? (projected changes in runoff?). L1: what is negatively affected by the increasing temperature? (annual runoff?). L6-7: temperature will decrease runoff. L17: temperature based PET tends to be oversensitive to changes in temperature compared to Penman-Monteith.*
**Author reply**
We will clarify these issues in the revised manuscript.

**Reviewer comment**

*P27 Table 1: Table 1 and Fig. 3 contain to a large degree redundant information. To me it is most important to have an impression of the general trends of the 5 climate variables T, P, Sh, Rs and Ws in the two RCPs while the origin country of a GCM is not relevant for the interpretation of the results. Since Fig. 3 provides the trend information of the climate variables, I recommend to delete Table 1 and list the names of the GCMs in the text of section 2.3.*

**Author reply**

Future changes in precipitation and temperature are the focus of this study, and the results support that these two variables are likely to have larger effects on runoff than other variables. Although we have showed the general trends of climate change in Fig. 3, it is difficult to see the magnitude of changes due to the large uncertainty ranges. Especially for precipitation, it is the most difficult variable to predict yet the most important input for runoff modeling. Our results directly depend on the projections of these GCMs. Therefore, before presenting the results, we want to give readers some idea about the magnitudes of climate change in the US and how different models agree or disagree on it.

**Reviewer comment**

*P31 Fig. 1: I am not sure if these two figures are necessary. Fig. 1a is only used in the context of the WaSSi model, where the individual land cover types are listed. Since the map is not further used in the results or discussion part I probably would remove it. Fig. 1b could maybe also be skipped - WRR names could be added to Table 2 and WRR IDs could be added to the maps of Fig.5 and Fig.8. Having the IDs directly in the maps would support the readability of the results where usually a reference to the WRR is made.*

**Author reply**

We agree that land cover distribution is not essential to the interpretation of this study. We thus will delete Fig. 1a. However, we feel that the figure of WRRs might be necessary, especially for readers who are not familiar with the hydrologic regions in the US. This map can directly display the locations of major rivers and drainage basins. Showing WRR names in a table may make it difficult to match names with locations.

**Reviewer comment**

*P36 Fig. 6: This is a more general comment on the use of WRRs and therefore also applies to Table 2 and the corresponding results parts. I wonder how much the averaged results on the level of WRR actually tell us? WRR can be considered as very large watersheds spanning a wide range of land cover types and hydroclimates. The runoff response of subbasins of a WRR to changes in climate variables can therefore be very diverse, which can be seen in Fig. 8. From a hydrological perspective it would be interesting to see exactly these relationships between changes in runoff response and hydroclimate, land cover, etc. Averaging the runoff response over a WRR makes conclusions about possible relationships difficult. In my opinion it would be worth to analyze the runoff response to changes in P and T in dependence of the hydroclimate (e.g. see studies of Coopersmith et al., 2014; Sawicz et al., 2014) or the Köppen Geiger climate zones.*

**Author reply**

We agree it is an interesting issue worthy of further investigation. As a matter of fact, we indeed have tried to sort the results by aridity (the ratio of precipitation to potential evapotranspiration) and land cover types across the country, but unfortunately we could not find any particular pattern

due to the large uncertainty in the magnitudes of climate change. We followed the common practice to present results by WRRs, which could provide some useful information for water resources managers and stakeholders.

**Reviewer comment**
*P38 Fig. 8: The information of Table 3 and Fig. 8 is very similar. Is it possible to combine the two? The fact that solar radiation, wind speed and specific humidity have little effect on changes in runoff response is already illustrated in Fig. 7 and therefore does not need to be repeated in Table 3. The areal proportions for precipitation and temperature as driving factors could be directly added to the maps in Fig. 8.*
**Author reply**
Table 3 is a summary of Fig. 5 (mean runoff change) and Fig. 8 (relative role of precipitation and temperature). We tried to display that runoff in precipitation-dominated watersheds are more likely to increase due to the widespread increase in both precipitation and humidity, while runoff in temperature-dominated watersheds may either increase (the combined effects of other factors exceed the temperature effect) or decrease (temperature effect exceeds the combined effects of all the other factors). Since the spatial patterns in Fig. 5 and Fig. 8 are different, putting everything in Fig. 8 might be confusing and misleading. However, we agree that Table 3 can be simplified. We will rephrase the caption and delete the results of radiation and wind (all zero).

**Reviewer comment**
*P8 L3-16: The terms "climate variables" and "driving factors" are used interchangeably as synonyms, which can be confusing. I recommend to use only one of the two terms.*
*P8 L7-8: I recommend to write ": : :independent effects E of each driving factor Ci..."*
*P8 L15: Based on equation 3 I assume that the contributions of the climate variables are quantified by the absolute relative weights.*
**Author reply**
We will rephrase this section according to the reviewers' comments.

**Reviewer comment**
*P9 L13-19: The first two sentences about sensitivity are to my perception not so relevant and could be deleted. I don't fully understand the last sentence - does pooling mean averaging of results?*
**Author reply**
These two sentences are meant to briefly explain why we used this approach and why it is more robust for examining the long-term patterns in an uncertain future. It might provide some useful information for readers who are interested in the method.
"Pooled" will be changed to "collected".

**Reviewer comment**
*P11 L2: I would not use abbreviations in the title.*
**Author reply**
We will change the acronyms in titles to the actual words.

**Reviewer comment**

*P20 L7-9: I think it is not necessary to mention in the conclusion that the Midwest has vast areas of croplands and grasslands, because this was not a major finding of the study.*

**Author reply**

We will delete this sentence from the Conclusions.

**Reviewer comment**

*P32 Fig. 2: The R-square values mentioned at P7 L16 could be added to the graph.*

**Author reply**

We will add it.

**Reviewer comment**

*P36 Fig. 6: The figure caption explains the elements of a boxplot. If you think this is needed you should also add the explanation in Fig. 7 to be consistent. Additionally, I recommend to use the same y-axis labels in the two figures.*

**Author reply**

We will add a statement in the caption of Fig. 7: "The format of the box-whisker plots is the same as that in Figure 6". The y-axis in Fig. 6 and Fig. 7 was set different because the variations at regional scale (Fig. 6) are clearly larger than that at CONUS scale (Fig. 7).

**Reviewer comment**

*Please use the HESS guidelines for all abbreviations and units. E.g. P33 Fig. 3: adapt units from W/m2 to W m-2.*

**Author reply**

It will be fixed.

**Reviewer comment**

According to the HESS guidelines, authors are encouraged to briefly describe the contribution of each co-author in a section called "author contributions".

**Author reply**

It will be added.

---

## Author Comment (AC2) · 28 Aug 2017

**Reviewer#2 - Finlayson**

**Reviewer comment**
*The overall academic content of this paper is sound, exploring the possible future runoff across the coterminous United States under conditions that may develop as predicted global climate changes unfold. However, my concerns with this paper relate to the way this material is presented. The authors appear to be unaware that they are writing to a global audience, and not to a group who, like themselves, are very familiar with the geography of the coterminous USA and with the systems used for identifying watersheds and location in the USA. I list below a series of points to illustrate my concerns.*
**Author reply**
We thank Dr. Finlayson for pointing out this important issue. We will add a section of "Study area" at the beginning of "2 Mehtods" to introduce the American hydrologic unit system and clarify the object of this study. The descriptions involving the geography of the conterminous US will be revised through the manuscript. Point-by-point responses are listed below.

**Reviewer comment**
*P 1 Lines 5-6: The use of the phrase "hydrologic paradigms" seems inappropriate here.*
*What is at issue here is the strength or intensity of different hydrological processes. Paradigms are something rather different.*
**Author reply**
We will change it to 'water balance', which is a more general term and may be more appropriate here.

**Reviewer comment**
*P 1 Line 7: "intensification of hydrologic cycle". What does this phrase mean?*
**Author reply**
We referred to the phenomenon that climate warming causes general increases in evaporation and precipitation, and higher frequency of extreme hydrologic events, which indicates an intensification (or acceleration) of the water cycle (Huntington, 2006).
*Huntington, Thomas G. "Evidence for intensification of the global water cycle: review and synthesis." Journal of Hydrology 319.1 (2006): 83-95.*

**Reviewer comment**
*P 1 Line 12: The use of "sustainably" in this context seems rather out of place. There are a lot of surface water sources and shallow aquifers that are being used very unsustainably.*
**Author reply**
This sentence will be revised to "runoff is a critical source of fresh water for humans".

**Reviewer comment**

*P 4 Lines 16-17 "the rate of decadal change of temperature over the CONUS has reached - 0.03~+0.28 °C since 1960s". I'm not sure what this means, it needs to be more clearly stated.*
**Author reply**
This sentence will be revised to "the rate of decadal change in temperature over the CONUS fluctuated between -0.03 °C and +0.28 °C from 1960s to 2000s".

**Reviewer comment**
*The authors assume that the readers have an intimate knowledge of some of the materials they are working with. So, for example, they use the term "8-digit Hydrologic Unit Code (HUC-8) watersheds" and "2-digit HUC Watershed". I have no idea what these are and I suspect I'm not the only one. The paper needs to be written for an international audience and not a just a group of those specialising in North American hydrology.*
**Author reply**
We will add a section "2.2.1 Study area" to clarify the object of this study and introduce the American hydrologic unit system.

**Reviewer comment**
*P 8 I do not follow the discussion from Line 3 to Line 17. Especially this term (Line 12) - R(C1t1,...,Ci t2 ,...,CNt1 ) - R(C1t1,...,Ci t1,...,CNt1 ). What is going on here needs to be explained more clearly, or is there a misprint?*
**Author reply**
$R(C1_{t1},\ldots,Ci_{t1},\ldots,CN_{t1})$ denotes runoff under the climate condition in the time period of *t1*. $R(C1_{t1},\ldots,Ci_{t2},\ldots,CN_{t1}) - R(C1_{t1},\ldots,Ci_{t1},\ldots,CN_{t1})$ denote runoff change driven by the change in variable *Ci* from *t1* to *t2*, while other variables remain constant.
  We will rephrase this section according to reviewers' comments.

**Reviewer comment**
*P 8 Line 20 "statistically downscaled" What does this mean? Is this a way of saying that the means or the medians were used?*
**Author reply**
We were trying to say that the data was corrected and downscaled from raw climate model outputs using statistical downscaling methods. This sentence will be broken into shorter sentences to avoid confusion.

**Reviewer comment**
*P 9 lines 1-2 "RCP4.5 and RCP8.5 were adopted as representatives of the intermediate and high emission scenarios respectively". At this point in the paper the readers have no idea what RCP4.5 and RCP8.5 are. There is some explanation later in the paragraph but it is not particularly clear. These terms need to be defined before they are used.*
**Author reply**
We will rewrite this paragraph to clarify the datasets and scenarios used in this study.

**Reviewer comment**

*Similarly, in Section 3, where the results are presented, Water Resource Regions (WRR) are referred to by their numbers and sometimes also the name of a general region, such as Midwest, Mountain West or coastal regions, in this case with no indication which bits of the coastal US are being referred to.*

**Author reply**

In the revised version, we will avoid terms that are not familiar to international audience, such as 'Midwest' and 'Mountain West'. We will use more general descriptions such as 'central U.S.' and 'northwest of U.S.'.

  In P14 L9, "coastal regions (WRR1,2,18)" will be revised to "Atlantic coast (WRR1,2) and Pacific coast (WRR18)".

**Reviewer comment**

*The writing style is rather unsatisfactory with frequent lack of the definite article and missing and incorrect words. Here is an example: "For example, slight decreases in P but somewhat increases in R are projected in south Texas due to the alteration of innerannual climate variability." I suspect that this, and the many similar cases in the text, come about from reviewing the text using the word processor's spelling check rather than careful reading by the authors.*

**Author reply**

We will recheck the writing more carefully based on the reviewers' comments.

**Reviewer comment**

*In Section 4.3 the authors argue that the results presented here indicate that "Additional water storage such as reservoirs and flood prevention measures may be needed in regions expecting more R". That may be the case but there is no evidence in this study that relates to flood behaviour and simply an increase in runoff does not say anything one way or the other about how floods will behave.*

**Author reply**

We will delete the statement about flood.

---

## Author Response (AR2)

**Comments**

*I thank the authors for answering the comments and implementing many suggestions. I am satisfied with the answers to my comments. I only have four short suggestions left that could be taken into account for the final manuscript.*

*P6 L9-10: I would explicitly state how many and which submodels build up WaSSI. E.g. the second and third sentence of the authors reply to comment 2 "WaSSi was integrated from a snow model, ...." would perfectly fit and definitively help guiding the reader through the modelling paragraph.*

*P6 L10: To get a better picture of the WaSSI model it would be helpful to know the total number of parameters that constitute the model.*

*P38 L8: I suggest to replace "... Figure 6" by "... Fig.6".*

*General comment: According to HESS guidelines a space must be included between numbers and units (change 10% to 10 %)*

**Response**

We want to thank the reviewer again for looking through the paper so carefully. We have corrected these issues.

[revised manuscript text omitted]